# High prevalence of plasmid-mediated quinolone resistance (PMQR) among *E. coli* from aquatic environments in Bangladesh

**Mohammed Badrul Amin**[1]*, **Sumita Rani Saha**[1], **Md Rayhanul Islam**[1], **S. M. Arefeen Haider**[1], **Muhammed Iqbal Hossain**[1], **A. S. M. Homaun Kabir Chowdhury**[1], **Emily K. Rousham**[2], **Mohammad Aminul Islam**[1,3]*

**1** Laboratory of Food Safety and One Health, Laboratory Sciences and Services Division, International Centre for Diarrhoeal Disease Research, Bangladesh (icddr,b), Dhaka, Bangladesh, **2** Centre for Global Health and Human Development, School of Sport, Exercise and Health Sciences, Loughborough University, Loughborough, United Kingdom, **3** Paul G. Allen School for Global Health, Washington State University, Pullman, Washington, United States of America

* badrul.amin@icddrb.org (MBA); amin.islam@wsu.edu (MAI)

**Data Availability Statement:** All relevant data are within the manuscript. In addition, details of isolates, genes and antibiotic susceptibility data are

## Abstract

Fluro(quinolones) is an important class of antibiotic used widely in both human and veterinary medicine. Resistance to fluro(quinolones) can be acquired by either chromosomal point mutations or plasmid-mediated quinolone resistance (PMQR). There is a lack of studies on the prevalence of PMQR in organisms from environmental sources in Bangladesh. In this study, we investigated the occurrence of PMQR genes in *E. coli* from various water sources and analysed associations between multi-drug resistance (MDR) and resistance to extended spectrum β-lactam antibiotics. We analysed 300 *E. coli* isolates from wastewaters of urban live-bird markets (n = 74) and rural households (n = 80), rural ponds (n = 71) and river water samples (n = 75) during 2017–2018. We isolated *E. coli* by filtering 100 ml of water samples through a 0.2μm cellulose membrane and incubating on mTEC agar media followed by identification of isolated colonies using biochemical tests. We selected one isolate per sample for detection of PMQR genes by multiplex PCR and tested for antibiotic susceptibility by disc diffusion. Clonal relatedness of PMQR-positive isolates was evaluated by enterobacterial repetitive intergenic consensus-PCR (ERIC-PCR). About 66% (n = 199) of *E. coli* isolates harbored PMQR-genes, predominantly *qnrS* (82%, n = 164) followed by *aac (6')-lb-cr* (9%, n = 17), *oqxAB* (7%, n = 13), *qnrB* (6%, n = 11) and *qepA* (4%, n = 8). Around 68% (n = 135) of PMQR-positive isolates were MDR and 92% (n = 183) were extended spectrum β-lactamase (ESBL)-producing of which the proportion of positive samples was 87% (n = 159) for $bla_{\text{CTX-M-1}}$, 34% (n = 62) for $bla_{\text{TEM}}$, 9% (n = 16) for $bla_{\text{OXA-1}}$, $bla_{\text{OXA-47}}$ and $bla_{\text{CMY-2}}$, and 2% (n = 4) for $bla_{\text{SHV}}$. Further, 16% (n = 32) of PMQR-positive isolates were resistant to carbapenems of which 20 isolates carried $bla_{\text{NDM-1}}$. Class 1 integron (*int*1) was found in 36% (n = 72) of PMQR-positive *E. coli* isolates. PMQR genes were significantly associated with ESBL phenotypes (*p*≤0.001). The presence of several PMQR genes were positively associated with ESBL and carbapenemase encoding genes such as *qnrS* with $bla_{\text{CTXM-1}}$ (*p*<0.001), *qnrB* with $bla_{\text{TEM}}$ (*p*<0.001) and $bla_{\text{OXA-1}}$ (*p* = 0.005), *oqxAB* and *aac (6')-lb-cr* with $bla_{\text{SHV}}$ and $bla_{\text{OXA-1}}$ (*p*<0.001), *qnrB* with $bla_{\text{NDM-1}}$ (*p*<0.001), *aac(6')-lb-cr* with

open access and available at the NERC Environmental Data Repository: https://doi.org/10.5285/0239cdaf-deab-4151-8f68-715063eaea45 and https://doi.org/10.5285/dda6dd55-f955-4dd5-bc03-b07cc8548a3d. The nucleotide sequence data of PMQR gene amplified fragments have been deposited in the PubMed GenBank nucleotide sequence database (http://www.ncbi.nlm.nih.gov/) and resulting GenBank under accession numbers were: qnrS (OL439745), qnrB (OL439744), oqxAB (OK668389), qepA (OK668390) and aac(6')-lb-cr (OL439743).

**Funding:** This research was funded by the Antimicrobial Resistance Cross Council Initiative supported by the seven research councils in partnership with the Department of Health and Department for Environment Food & Rural Affairs (NERC/ BBSRC/MRC grant number: NE/N019555/1). Dr. Emily K. Rousham received this grant.

**Competing interests:** The authors have declared that no competing interests exist.

$bla_{OXA-47}$ ($p<0.001$) and $bla_{NDM-1}$ ($p = 0.002$). Further, *int*1 was found to correlate with *qnrB* ($p<0.001$) and *qepA* ($p = 0.011$). ERIC-PCR profiles allowed identification of 84 of 199 isolates with 85% matching profiles which were further grouped into 33 clusters. Only 5 clusters had isolates (n = 11) with identical ERIC-PCR profiles suggesting that PMQR-positive *E. coli* isolates are genetically heterogeneous. Overall, PMQR-positive MDR *E. coli* were widely distributed in aquatic environments of Bangladesh indicating poor wastewater treatment and highlighting the risk of transmission to humans and animals.

## Introduction

With 700,000 global deaths annually, bacterial infections caused by antimicrobial-resistant organisms are a major public health concern [1]. Antimicrobial-resistant (AMR) infections increase mortality, treatment duration, recovery time, and health care costs. AMR is a One Health problem and addressing the issues related to human and animal health separately or in combination is not enough if the environmental dimensions of the problem are not addressed. The emergence of multidrug-resistant (MDR) organisms in aquatic environments constitutes a major threat for both humans and livestock. Although *E. coli* is part of the normal flora in humans and animals, pathogenic strains of *E. coli* cause severe clinical challenges including gastrointestinal tract infection, central nervous system infection, urinary tract and skin and soft tissue infections [2, 3]. These infections become more severe when caused by MDR pathogens [4]. Several recent investigations reported the emergence of MDR bacteria from different host origins including humans, birds, cattle, and fish that increase the need for antimicrobial susceptibility testing to identify the antibiotic of choice as well as screening for emerging MDR strains [5–11].

Quinolones and fluoroquinolones (FQs) are broad-spectrum antibiotics frequently used in human and veterinary health for treatment of both Gram-positive and Gram-negative bacterial infections [12]. Fluro(quinolones) are the third most commonly prescribed antibiotics in Bangladesh for treating outpatients suffering from common cold and fever, infections, diarrhea, and gonorrhea [13, 14]. In livestock production in Bangladesh, flouroquinolone is one of the most commonly used antibiotics, with an estimated consumption of 100 metric ton per year [15]. Ciprofloxacin is widely used as a single drug or in combination with other drugs and is often sold as feed supplements under many different brand names [15]. Farmers use this antibiotic mostly for prophylactic purposes to avert infections or as an alternative to good agricultural practices and as a growth promoter on farm animals [16, 17]. With increasing use of fluoroquinolones, the prevalence of fluoroquinolone resistance has also been increasing. Although quinolone resistance in *Enterobacteriaceae* is mainly attributed to point mutations in quinolone resistance-determining regions (QRDRs) of the type II topoisomerase genes (gyrase: *gyrA, gyrB*; and topoisomerase IV: *parC, parE*), there are an increasing number of reports of plasmid-mediated quinolone resistance (PMQR) determinants associated with low-level resistance to fluoroquinolones [18–21]. Moreover, bacterial pathogens that are positive for PMQR are also more likely to have chromosomal mutations resulting in high level of resistance to the antibiotics [22, 23]. Co-occurrence of PMQR with extended spectrum β-lactamase (ESBL) genes may limit the treatment options for infections caused by ESBL-producing bacteria [24].

Three categories of PMQR genes have been reported based on their mode of action. Such examples include the *qnr* alleles (*qnrA, qnrB, qnrS, qnrC,* and *qnrD)*; efflux pump genes (e.g.

*oqxAB*, *qepA*); and a variant of aminoglycoside acetyl transferase (*aac-(6′)-Ib-cr*) [25–28]. *qnrA*, *qnrB* genes can be carried by large and usually conjugative plasmids, whereas *qnrS* can be carried by small, mobilizable, and non-conjugative plasmids [18, 29]. However, both types of plasmid can readily disseminate and transmit antibiotic resistance traits among bacterial communities. Another important mechanism of PMQR gene transmission among the bacterial population is via the integrons (*int*) particularly *int*1 in Gram-negative bacteria [30, 31]. Close proximity between antibiotic resistance genes and *int*1 thus enhances mobility by transposition and allows them to become associated with multiple antibiotic-resistant gene (ARG) cassettes and heavy metal and disinfectant resistant genes [18, 32].

The prevalence of PMQR genes has been investigated in different countries across the world. Previous studies in humans, food-producing animals, wild animals, and wastewater samples showed an overall prevalence of PMQR in *E. coli* of 25% with the highest reported occurrence (49%) in retail turkey from Czech Republic [33]. In China aquatic environmental samples had a 30% prevalence of PMQR-positive *E. coli* isolates overall; with a prevalence of 28% in hospital-impacted water samples and 37% in aquaculture-impacted river water samples [34]. Limited information is available on the prevalence of PMQR in Bangladesh. PMQR genes were detected in clinical isolates of *E. coli* and *K. pneumoniae* largely from wound and urinary tract infections, and in *E. coli* from cloacal swabs of poultry [35, 36]. Recently, a novel quinolone resistance gene *qepA* has been detected in *E. coli* and *K. pneumoniae* strains isolated from lake and river water samples in Bangladesh [37]. Although previous studies have shown that aquatic environments in Bangladesh particularly drinking water, wastewater, and surface water bodies such as ponds and rivers are heavily contaminated with various faecal pathogens including multi-drug resistant organisms, no studies have estimated the prevalence of PMQR among isolates [38, 39]. In this study, we aimed to investigate water samples from different aquatic environments including wastewater and surface water from both rural and urban areas of Bangladesh to understand the prevalence and distribution of *E. coli* carrying PMQR along with their resistance patterns against clinically important antibiotics. A further aim was to investigate the association of PMQR genes with ESBL- and carbapenemase-producing genes in environmental *E. coli* isolates.

## Methods and materials

### Ethical approval

This research protocol was approved by the Institutional Review Board of the International Centre for Diarrhoeal Disease Research, Bangladesh (icddr,b) (protocol number PR-16071).

### Study overview

The present study is part of a larger study that simultaneously examined the dynamics of AMR transmission from contaminated outdoor environments such as poultry, soil, surface water, solid waste, and wastewater from urban and rural Bangladesh [40]. Here, we investigated the *E. coli* isolates collected from wastewater of rural poultry farms and households, and urban live-bird markets, as well as pond water and river water samples from rural areas to determine the presence of PMQR genes and analyse their distribution according to sample types.

### Sample collection

A total of 300 water samples including urban wastewater (n = 74), rural wastewater (n = 80) from poultry farms and households, river (n = 75) and pond (n = 71) water samples were collected during 2017–18 following previously described procedures [40]. Briefly, water samples

were collected using a sterile plastic bottle filled by plunging downwards about 30 cm below the water surface. Sample bottles were placed in a cool box (4–8˚C) and transported to the laboratory within 8 hours of collection for culture.

## Isolation and identification of *E. coli*

About 100 mL of water sample was passed through a 0.2 μm cellulose nitrate filter (Sartorius Stedim Biotech GmbH, Goettingen, Germany) and then the filter was placed in an upright position on modified mTEC agar media (BD Difco, New Jersey, USA). The culture plate was incubated at 37˚C for 2 hours and then at 44˚C overnight to allow growth of thermotolerant *E. coli*. mTEC medium contains a chromogen (5-bromo-6-chloro-3-indolyl-β-D-glucuronide), which is catabolized by *E. coli* producing β-D-glucuronidase to glucuronic acid and produces a red- or magenta-coloured compound. Two *E. coli* isolates were selected from each water sample and sub-cultured on MacConkey agar (BD Difco, New Jersey, USA) and incubated at 37˚C for overnight. The presumptive colonies were identified according to their colony characters, microscopical examination using Gram staining, motility test, and biochemical reactions (oxidase, catalase, indole, lactose fermentation, methyl-red, citrate-utilization, $H_2S$, Voges-Proskauer, and urease tests) as described previously [41]. All *E. coli* isolates were stored in Tryptone soya broth (Oxoid Limited, Hampshire, England) with 30% glycerol (Sigma-Aldrich, Darmstadt, Germany) and kept at -80˚C for future use.

## Screening of PMQR-positive isolates by PCR

All *E. coli* isolates from water samples were investigated for plasmid-mediated quinolone resistance genes (*qnrS*, *qnrB*, *oqxAB*, *qepA*, *aac(6')-lb-cr*, *qnrA*, *qnrC* & *qnrD*) by multiplex-PCR as described previously [42]. The primers and PCR cycling are listed in Table 1.

## Confirmation of amplified fragments of PMQR genes by sequencing

The PCR amplified fragment of each PMQR gene found in this study was sequenced using ABI PRISM BigDye Terminator Cycle Sequencing Reaction kit (Applied Biosystems; CA,

**Table 1. List of primer sequences used in multiplex PCR for the determination of PMQR determinants.**

| Target gene | Primer sequences | Amplicon size (bp) | Amplification (30 cycles) | | | References |
|---|---|---|---|---|---|---|
| | | | Denaturation | Annealing | Extension | |
| *qnrA* F | CAGCAAGAGGATTTCTCACG | 630 | 94˚C for 30 seconds | 63˚C for 90 seconds | 72˚C for 90 seconds | Ciesielczuk et al., 2013 [42] |
| *qnrA* R | AATCCGGCAGCACTATTACTC | | | | | |
| *qnrD* F | CGAGATCAATTTACGGGGAATA | 581 | | | | |
| *qnrD* R | AACAAGCTGAAGCGCCTG | | | | | |
| *qnrB* F | GGCTGTCAGTTCTATGATCG | 488 | | | | |
| *qnrB* R | GAGCAACGATGCCTGGTAG | | | | | |
| *qnrS* F | GCAAGTTCATTGAACAGGGT | 428 | | | | |
| *qnrS* R | TCTAAACCGTCGAGTTCGGCG | | | | | |
| *oqxAB* F | CCGCACCGATAAATTAGTCC | 313 | | | | |
| *oqxAB* R | GGCGAGGTTTTGATAGTGGA | | | | | |
| *aac(6')-lb-cr* F | TTGGAAGCGGGGACGGAM | 260 | | | | |
| *aac(6')-lb-cr* R | ACACGGCTGGACCATA | | | | | |
| *qepA* F | GCAGGTCCAGCAGCGGGTAG | 218 | | | | |
| *qepA* R | CTTCCTGCCCGAGTATCGTG | | | | | |
| *qnrC* F | GCAGAATTCAGGGGTGTGAT | 118 | | | | |
| *qnrC* R | AACTGCTCCAAAAGCTGCTC | | | | | |

USA) and ABI PRISM 310 automated sequencer (Applied Biosystems; CA, USA). Before that, PCR products were purified using the PCR Purification kit (QIAGEN, Hilden, Germany) according to the manufacturer's instructions. BioEdit software was used to analyse the raw sequence reads and for determining the homology with deduced sequence size. All the PMQR gene sequences were searched and confirmed by Basic Local Alignment Search Tool (BLAST). Finally, the sequences were submitted to GeneBank under five accession numbers: *qnrS* (OL439745), *qnrB* (OL439744), *oqxAB* (OK668389), *qepA* (OK668390) and *aac(6')-lb-cr* (OL439743).

## Antibiotic susceptibility testing of PMQR-positive *E. coli*

All PMQR-positive *E. coli* isolates were tested for antibiotic susceptibility against 16 clinically important antibiotic agents under nine antibiotic classes by disc diffusion method [43]. Commercially available antibiotic discs (Oxoid Limited, Hampshire, England) used for the test were: ampicillin (10 μg), cefotaxime (30 μg), ceftriaxone (30 μg), ceftazidime (30 μg), cefixime (5 μg), cefepime (30 μg),cefoxitin (30 μg), ciprofloxacin (5 μg), nalidixic acid (30 μg), sulfamethoxazole/trimethoprim (25 μg), gentamycin (10 μg), nitrofurantoin(300 μg), imipenem (10 μg), meropenem (10 μg), ertapenem (10 μg) and piperacillin-Tazobactam (10 μg). Results were interpreted as sensitive and resistant, and isolates showing resistance against at least one agent of more than three classes of antibiotics were classified as MDR. Where there was resistance against at least one agent in all classes except two or fewer antibiotic classes were considered as extensively drug-resistant (XDR) [43, 44]. Extended spectrum β-lactamase (ESBL) production was determined by using the combination disk test where a β-lactam inhibitor, clavulanic acid (30/10 μg) with cefotaxime (30 μg) and ceftazidime (30 μg) were used [43].

## Detection of ESBL, carbapenemase and integrase encoding genes

All ESBL-producing isolates were screened for $bla_{CTX-M-1}$, $bla_{CMY-2}$, $bla_{TEM}$, $bla_{SHV}$, $bla_{OXA-1}$, and $bla_{OXA-47}$ genes by PCR [45]. All carbapenem resistant isolates were tested for carbapenem resistance genes, $bla_{NDM-1}$ and $bla_{OXA-48}$ according to the procedure described earlier [46]. Class 1 integrons were detected by PCR for *int*1 gene using the primer sequences and PCR conditions as described earlier [47].

## Statistical analysis

Data were entered and analysed using Stata (Version 13.0, StataCorp LLC, College Station, TX, USA). Univariate analyses were performed to examine the presence of PMQR genes in different aquatic sources including wastewater, pond water and river water. In bivariate analyses, the prevalence of PMQR genes in different categories of antibiotic resistant *E. coli* isolates such as MDR, ESBL- and carbapenemase-producers was compared using *Chi*-square or Fisher's exact test with Bonferroni correction [48]. Correlations for binary variables were calculated using the 'cor' function and using 'cor.test' function in R software (version 4.0.2; https://www.r-project.org/). Significant correlations were visualized utilizing the 'corrplot' function from the 'corrplot' R package [49]. For all analyses, statistical significance was considered as $p < 0.05$.

## Phylogenetic analysis using ERIC-PCR

PMQR-positive *E. coli* isolates were analysed for clonal diversity using enterobacterial repetitive intergenic consensus-PCR (ERIC-PCR). Two primers, ERIC1 (5'-ATGTAAGCTCC TGGGGATTCAC-3') and ERIC2 (5'-AAGTAAGTGACTGGGGTGAGCG-3') were used following previously reported procedures [50]. ERIC-PCR amplified products were then

separated in 1.5% agarose gel, normalised using the 100 bp DNA ladder as an external reference standard, stained with Midori Green, and visualized by FastGene Blue/Green LED Gel Illuminator (Nippon Genetics, Tokyo, Japan). The TIF formatted image was analysed with BioNumerics version 4.5 (Applied Maths, Kortrijk, Belgium) to determine phylogenetic similarity among the isolates. Dendrogram clusters based on dice and clustering correlation coefficient showing 85% similarity in banding patterns between *E. coli* isolates were considered as phylogenetically related [51].

## Results

### Phenotypic characteristics of *E. coli* isolated from water samples

Isolates were identified as *E. coli* based on their morphology and biochemical characteristics. Microscopically, the bacteria appeared as Gram-negative moderate size, motile, and non-sporulated rods. The bacteria grew well on mTEC agar and appeared as red/magenta colour colonies due to β-D-glucuronidase activity which is highly specific for *E. coli*. On MacConkey agar, bacteria produced characteristic pink colonies due to lactose fermentation. Biochemically, all isolates were positive for catalase, lactose fermentation, indole and methyl-red, tests. Simultaneously, they were negative for cytochrome oxidase, Voges-Proskauer, citrate-utilization, H2S production, and urease tests. All 300 water samples were positive for *E. coli*.

### PMQR genes are highly prevalent in *E. coli* from water sources

Of 300 *E. coli* isolates from water samples, 66% (n = 199) were positive for PMQR genes (Table 2). Of these, the majority of isolates were positive for *qnrS* (82%, n = 164) followed by *aac(6')-lb-cr* (9% n = 17), *oqxAB* (7%, n = 13) *qnrB* (6%, n = 11) and *qepA* (4%, n = 8). None of the isolates was positive for *qnrA*, *qnrC* & *qnrD*. *qnrS* was predominantly detected in rural pond water (90%, n = 45) whereas *aac(6')-lb-cr* was found in river water (24%, n = 11). Ten *E. coli* isolates carried more than one PMQR gene in the following combinations: *qnrS+oqxAB* (n = 2), *qnrS+qnrB+oqxAB* (n = 2), *qnrB+oqxAB+aac(6)'-lb-cr* (n = 1), *qnrS+qnrB+aac(6)'-lb-cr* (n = 1), *qnrS+qepA+oqxAB* (n = 1), *qnrS+qnrB* (n = 1), *qnrS+oqxAB* (n = 1) and *qnrB +oqxAB* (n = 1).

### PMQR-positive *E. coli* are predominantly multi-drug resistant

All PMQR-positive isolates (n = 199) were resistant to penicillin followed by 96% resistant to cephamycins and extended spectrum cephalosporins, 48% to fluro(quinolones), 32% to folate pathway inhibitors, 26% to nitrofuran, 23% to β-lactamase inhibitors, 21% to aminoglycoside and 16% resistant to carbapenem (Table 3). Further, the distribution of antibiotic resistance patterns of PMQR positive isolates was analysed according to their sources. Of 199 PMQR

**Table 2. Prevalence of PMQR genes in *E. coli* isolates obtained from different aquatic environments.**

| Sampling site | Sample Type | Sample number (n) | PMQR positive n (%) | No. (%) of isolates positive for PMQR genes | | | | |
|---|---|---|---|---|---|---|---|---|
| | | | | *qnrS* n (%) | *qnrB* n (%) | *oqxAB* n (%) | *qepA* n (%) | *aac(6')-lb-cr* n (%) |
| Urban | Wastewater | 74 | 45 (61) | 52 (65) | 3 (4) | 4 (5) | 0 | 2 (3) |
| Rural | Wastewater | 80 | 58 (72) | 37 (50) | 4 (5) | 3 (4) | 3 (4) | 4 (5) |
| | pond water | 71 | 50 (70) | 45 (90) | 2 (4) | 3 (6) | 2 (4) | 0 |
| | river water | 75 | 46 (61) | 30 (65) | 2 (4) | 3 (7) | 3 (7) | 11 (24) |
| | Total | 300 | 199 (66) | 164 (82) | 11 (6) | 13 (7) | 8 (4) | 17 (9) |

*R, resistance; n, number.

**Table 3. Occurrence of clinically important antibiotic resistance among PMQR-positive *E. coli* isolates from aquatic environments in Bangladesh.**

| Antibiotic classes | Antibiotics tested | No. (%) of *E. coli* resistant | | | | p value |
|---|---|---|---|---|---|---|
| | | Urban Wastewater (n = 45) | Rural Wastewater (n = 58) | Rural pond water (n = 50) | Rural river water (n = 46) | |
| Aminoglycoside | Gentamycin | 18 (40) | 7 (12) | 2 (4) | 15 (33) | $p<0.05$ |
| Antipseudomonal penicillins plus β-lactamase inhibitors | Piperacillin-Tazobactam | 10 (22) | 8 (14) | 6 (12) | 21 (46) | $p<0.001$ |
| Cephamycins | Cefoxitin | 44 (98) | 56 (97) | 47 (94) | 44 (96) | 0.654 |
| Extended-spectrum cephalosporins | Cefotaxime | 44 (98) | 56 (97) | 45 (90) | 44 (96) | 0.165 |
| | Ceftriaxone | 44 (98) | 56 (97) | 45 (90) | 44 (96) | 0.165 |
| | Ceftazidime | 39 (87) | 47 (81) | 38 (76) | 43 (98) | 0.065 |
| | Cefixime | 43 (96) | 56 (97) | 47 (94) | 44 (96) | 0.838 |
| | Cefepime | 44 (98) | 55 (95) | 45 (90) | 44 (96) | 0.277 |
| Carbapenem | Ertapenem | 9 (20) | 5 (9) | 6 (12) | 10 (22) | 0.343 |
| | Meropenem | 10 (22) | 6 (10) | 7 (14) | 9 (19) | 0.363 |
| | Imipenem | 11 (24) | 8 (14) | 6 (12) | 7 (15) | 0.209 |
| Fluro(quinolone) | Nalidixic acid | 34 (76) | 23 (40) | 19 (38) | 27 (59) | 0.074 |
| | Ciprofloxacin | 34 (76) | 20 (34) | 14 (28) | 26 (57) | $p<0.05$ |
| Folate pathway inhibitors | Sulfamethoxazole/trimethoprim | 27 (60) | 15 (26) | 11 (22) | 11 (24) | $p<0.05$ |
| Nitrofuran | Nitrofurantoin | 9 (20) | 8 (14) | 15 (30) | 20 (43) | $p<0.05$ |
| Penicillin | Ampicillin | 45 (100) | 58 (100) | 50 (100) | 46 (100) | NA |
| MDR (≥3 Ab classes) | | 39 (87) | 32 (55) | 29 (58) | 35 (76) | 0.157 |
| XDR (All antibiotic classes except two or fewer classes) | | 8 (18) | 4 (7) | 2 (4) | 13 (28) | $p<0.05$ |

positive *E. coli* isolates, 68% (n = 135) and 14% (n = 27) were MDR and XDR respectively, of which urban wastewater and rural river water showed high abundance. Aminoglycosides, fluoroquinolones, nitrofuran and folate pathway inhibitor drug resistant isolates had significant relationship with their sources ($p<0.05$) (Table 3).

## A significant proportion of PMQR-positive *E. coli* were positive for ESBL genes

About 92% (n = 183) of PMQR-positive *E. coli* were ESBL-producers. Screening of ESBL encoding genes showed that 87% (n = 159) of isolates were positive for $bla_{CTX-M-1,}$ 34% (n = 62) for $bla_{TEM}$, 9% (n = 17) for $bla_{OXA-1}$, $bla_{OXA-47}$ and $bla_{CMY-2}$ each, and 2% (n = 4) for $bla_{SHV}$. Among carbapenem resistance, only $bla_{NDM-1}$ was detected in 10% (n = 20) of the PMQR-positive isolates. None of the isolates were positive for $bla_{OXA-48.}$ Further, the class 1 integron encoding gene, *int*1, was detected in 37% (n = 73) of the isolates.

Presence of PMQR genes was significantly associated with the ESBL phenotype ($p<0.001$) of the isolates. At the gene level, $bla_{CTX-M-1}$ ($p<0.001$) and $bla_{TEM}$ ($p<0.05$) were significantly more common in PMQR-positive versus negative isolates (Table 4). A correlation matrix analysis between presence of PMQR genes and presence of ESBL, carbapenemase and class 1 integron encoding genes was conducted (Fig 1). The presence of *qnrS* was positively associated with $bla_{CTXM-1}$ ($p<0.001$) while *qnrB* was positively associated with $bla_{TEM}$ ($p<0.001$), $bla_{OXA-1}$ ($p = 0.005$) and $bla_{NDM-1}$ ($p<0.001$). Detection of PMQR genes *oqxAB* and *aac(6')-lb-cr* in isolates was positively associated with ESBL genes $bla_{SHV}$ and $bla_{OXA-1}$ ($p<0.001$ for both). Isolates positive for the PMQR efflux gene, *qepA*, were predominantly positive for AmpC β-lactamase gene $bla_{CMY}$ ($p = 0.016$). Isolates carrying *aac(6')-lb-cr* gene were more likely to be

**Table 4. Association of PMQR genes with ESBL, carbapenemase, integrase genes, and MDR and XDR phenotypes in *E. coli* isolates.**

| Characteristics | No. (%) of *E. coli* | | *p* value[*] |
|---|---|---|---|
| | **PMQR positive (n = 199)** | **PMQR negative (n = 101)** | |
| ESBL | 183 (92) | 74 (73) | p<0.001 |
| $bla_{CTXM-1}$ | 165 (83) | 51 (50) | p<0.001 |
| $bla_{SHV}$ | 5 (3) | 1 (1) | 0.668 |
| $bla_{TEM}$ | 63 (32) | 46 (46) | p<0.05 |
| $bla_{OXA-1}$ | 17 (9) | 12 (12) | 0.355 |
| $bla_{OXA-47}$ | 16 (8) | 13 (13) | 0.181 |
| Carbapenemase | 32 (16) | 17 (17) | 0.70 |
| $bla_{NDM-1}$ | 20 (10) | 16 (16) | 0.145 |
| $bla_{CMY-2}$ | 16 (8) | 8 (8) | 0.971 |
| $bla_{OXA-48}$ | 0 | 0 | NA |
| Class 1 integron (*int*1) | 72 (36) | 40 (40) | 0.562 |
| MDR | 135(68) | 74 (73) | 0.334 |
| XDR | 27 (14) | 20 (20) | 0.160 |

[*]*p* values were determined using *Chi*-square or Fisher's exact test.

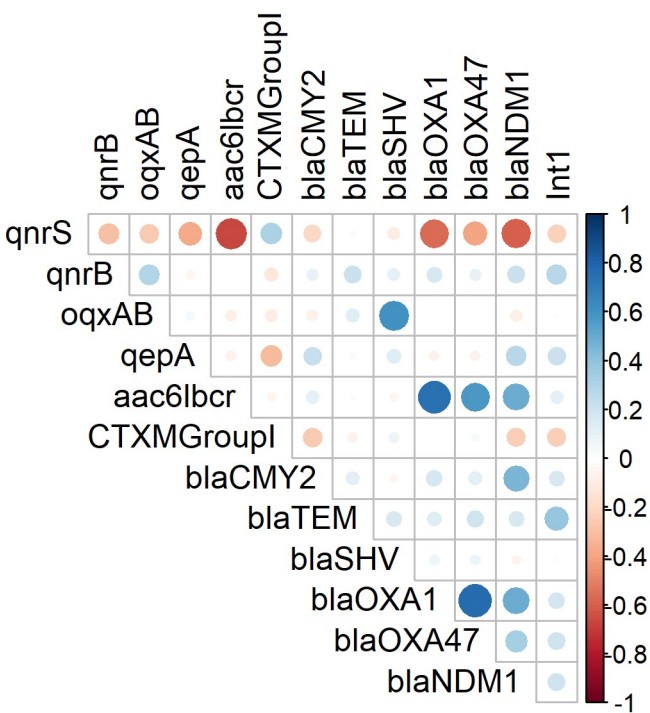

**Fig 1. Correlation matrix of the presence of PMQR genes with ESBL and carbapenemase and integron (*int*1) encoding genes in *E. coli*.** White spaces are not significantly correlated. Blue circles indicated significant positive correlation and red showed significant negative correlation. The size and strength of colour represent the numerical value of the Phi correlation coefficient.

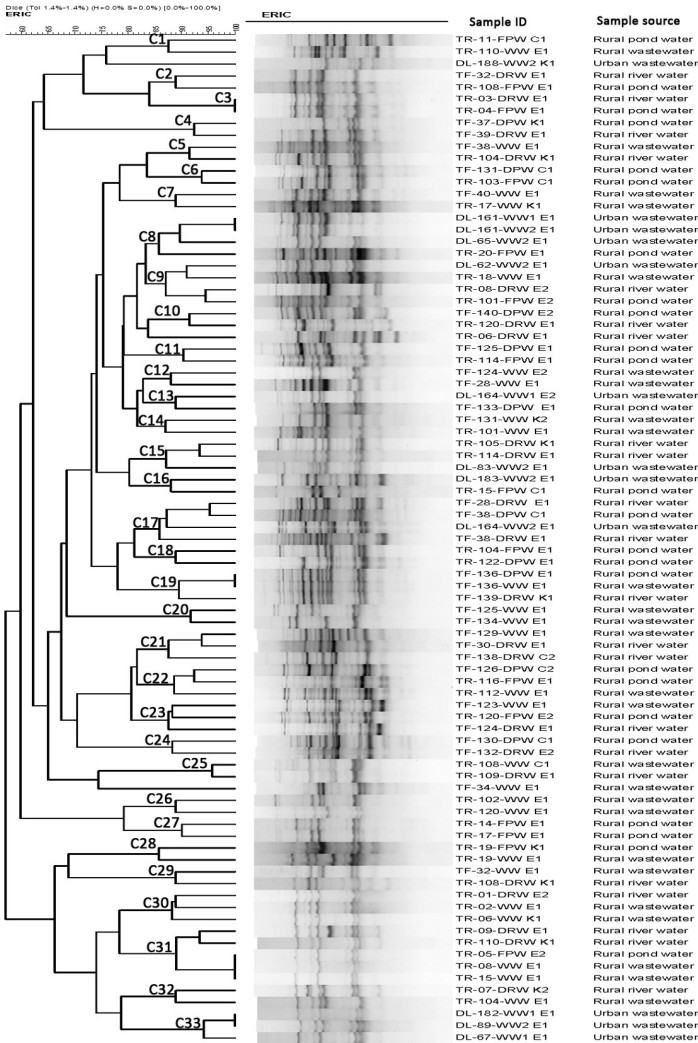

**Fig 2. Dendrogram generated by BioNumerics software, showing distances calculated by the Dice similarity index of ERIC-PCR banding patterns of *E. coli* strains isolated from various aquatic samples.** The degree of similarity (%) is shown on the scale. The isolates were considered as phylogenetically related based on 85% similarity in ERIC-PCR banding patterns.

positive for carbapenem resistance genes $bla_{OXA-47}$ ($p<0.001$) and $bla_{NDM-1}$ ($p = 0.002$). Unlike antibiotic resistance genes, the presence of Class 1 integron gene *Int*1 was associated with a diverse group of PMQR genes including *qnrB* ($p<0.001$) and *qepA* ($p = 0.011$).

## ERIC-PCR analysis

According to the 85% cut-off for similarity in ERIC-PCR banding patterns, 84 of 199 (42%) PMQR-positive *E. coli* isolates were grouped into 33 clusters (C1-C33) and the isolates in different clusters were randomly distributed irrespective of the sources of isolation (Fig 2). Among these 84 *E. coli* isolates, only 11 in five clusters, C3, C8, C19, C31 and C33, had identical banding patterns indicating clonal relationships. *E. coli* isolates from wastewater samples from different locations in urban areas belonged to the same clusters (C8 and C33) whereas isolates from pond and river water samples in rural areas belonged to the same clusters (C3).

Further, isolates from pond and river water were grouped in clusters C27 and C46, respectively with isolates from wastewater samples in rural areas.

## Discussion

In Bangladesh, fluro(quinolones) is one of the most frequently used antibiotic classes in both human and veterinary medicine although the major usage is in animal husbandry where it is applied as a feed supplement for prophylaxis and growth promotion [13–16]. A recent report indicates that few farmers use ciprofloxacin for bacterial disease prevention in aquaculture in Bangladesh [52]. Aquatic environments are more likely, therefore, to be contaminated with residual fluro(quinolones) via effluents from both human and animal wastes which may be contributing to the high prevalence of fluoroquinolone resistance in bacterial organisms. In particular, PMQR genes along with ESBL and carbapenemase encoding genes are likely to be carried by the same plasmids that can be shared with other organisms in aquatic environments by horizontal transmission. Except for one investigation of 12 *E. coli* isolates from water samples reporting 4 positives for *qnrS*, no other study has characterised environmental isolates for PMQR in Bangladesh [37]. In this study, we found that a significant proportion (66%) of water samples were positive for *E. coli* carrying PMQR genes, predominantly *qnrS*. This percentage would have been greater if more than one isolate per sample had been selected. Interestingly, previous studies reported a higher prevalence of *qnrS* in *E. coli* isolates from patients with extraintestinal infections, and from poultry cloacal samples in Bangladesh [35, 36]. It is likely that bacterial isolates harboring *qnrS* from both clinical and poultry sources might be released to the water bodies through human or animal waste due to lack of proper wastewater treatment facilities in Bangladesh [53]. In contrast to our study, studies in other countries such as China, Switzerland, and Poland showed that *aac(6')-lb-cr* was the predominant PMQR gene in *E. coli* isolates from aquatic samples [34, 54–56]. Further investigations of the isolates using a comparative genomic approach will provide more insights into the characteristics of isolates driving this discriminate distribution of PMQR genes.

In the present study, more than 65% of the PMQR positive isolates were resistant to multiple antibiotics including penicillin, cephalosporins, fluro(quinolones), sulfonamides aminoglycosides and carbapenems. A high prevalence of MDR could be associated with the wide range of antibiotics used in the poultry and aquaculture sectors which increase selection pressures for AMR in water bodies. Different mechanisms can be involved in the emergence of MDR *E. coli* strains such as: 1) shared resistance mechanisms that occur for the antimicrobial agents in the same category, e.g., mutations in penicillin-binding protein and presence of the ß-lactamases. This can also occur for antibiotics in different classes due to the presence of efflux pumps acting on different antibiotics; 2) exposure to multiple antibiotics via routine use of combination therapy and repeated treatment failure and 3) the presence of plasmids that carry resistance genes to multiple antibiotics. We found that the presence of PMQR genes, particularly *qnr*, in *E. coli* isolates was associated with the ESBL phenotype (p<0.001) and various β-lactamase encoding genes including $bla_{CMY}$, $bla_{CTX-M-1}$, $bla_{CMY}$, $bla_{TEM}$ and $bla_{OXA-1}$. This can be explained by carriage of both *qnr* and ESBL/AmpC genes in the same plasmids as reported by previous studies [57–59]. Apart from *qnr*, other PMQR genes such as *oqxAB* were found to associate significantly with $bla_{SHV}$ whereas *aac(6')-lb-cr* was associated with $bla_{OXA-1}$ and $bla_{OXA-47}$. This finding concurs with earlier reports that indicated the co-occurrence of *oqxAB* with $bla_{SHV}$ and *aac(6')-lb-cr* with $bla_{OXA-1}$ and $bla_{OXA-47}$ [59, 60]. The presence of both PMQR and ESBL genes in the same bacterial isolates could be due to the co-selection of isolates in the environment with either of fluoroquinolone or cephalosporins which accentuate

further confirmation. Extensive use of quinolones therefore may lead to the emergence of resistance against β-lactams which are important clinically used antibiotics.

Integrons are important mobile genetic elements that carry different antibiotic resistance gene cassettes and play a crucial role in AMR transmission via horizontal gene transfer between different bacterial species [61]. Class 1 integron (*int*1) is most studied and reported ubiquitously in different enterobacterial species including *E. coli* [62]. In our study, *qnrB* and *qepA* were associated with the presence of *int*1 (Fig 1) indicating that these genes might be in the gene cassette carried by the integrons. Previous study reported that *int*1 in *E. coli* carried gene cassettes encoding resistance to multiple antibiotics including β-lactams (*bla*$_{OXA-30}$), trimethoprim (*dfrA1*, *dfrA5*, *dfrA7*, *dfrA12*, *dfrA17*), aminoglycosides (*aadA1*, *aadA2*, *aadA5*), chloramphenicol (*cmlA*) and erythromycin (*ereA2*) [63]. Findings from our study highlight the need for further investigation to identify whether PMQR genes are located in the class 1 integron of the isolates using whole genome sequencing.

In this work, ERIC-PCR analysis revealed that the PMQR-positive isolates were mostly heterogeneous although a small number of isolates from different sources of water or locations had identical banding patterns indicating their clonal relationship. It could be that certain clonal groups of PMQR-positive ESBL-producing *E. coli* were predominantly present in the waterbodies, however, further characterization of these clones using next generation sequencing would be useful. Further, comparison of these isolates with clinical isolates could be done to understand the contribution of these widely circulating clones on the burden of antimicrobial resistant infections in the community.

## Conclusions

Our study shows a high prevalence of PMQR-positive *E. coli* in urban and rural waters. These plasmid-mediated isolates were mostly MDR, predominantly ESBL-producing and genetically heterogeneous. The high prevalence of plasmid-mediated quinolone resistance poses a risk for horizontal gene transfer and this, in association with ESBL genes, adds to the threat of AMR transmission via the environment. This study highlights the importance of including surface waters and wastewaters in One Health AMR surveillance programs to understand the emergence and transmission dynamics of AMR and for designing environmental intervention strategies.

## Acknowledgments

icddr,b is grateful to the Governments of Bangladesh, Canada, Sweden and the United Kingdom for providing core/unrestricted support.

## Author Contributions

**Conceptualization:** Mohammed Badrul Amin, Mohammad Aminul Islam.

**Formal analysis:** Mohammed Badrul Amin, Md Rayhanul Islam.

**Funding acquisition:** Emily K. Rousham.

**Investigation:** Sumita Rani Saha, S. M. Arefeen Haider, Muhammed Iqbal Hossain, A. S. M. Homaun Kabir Chowdhury, Mohammad Aminul Islam.

**Methodology:** Mohammed Badrul Amin, Sumita Rani Saha.

**Project administration:** Emily K. Rousham, Mohammad Aminul Islam.

**Supervision:** Mohammed Badrul Amin.

**Writing – original draft:** Mohammed Badrul Amin.

**Writing – review & editing:** Emily K. Rousham, Mohammad Aminul Islam.

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
