## [Decision Letter · Decision Letter 0]

25 Oct 2021

PONE-D-21-32725Prevalence and characterization of plasmid mediated quinolone resistance (PMQR) in E. coli isolated from aquatic environments of BangladeshPLOS ONE

Dear Dr. Amin,

Thank you for submitting your manuscript to PLOS ONE. After careful consideration, we feel that it has merit but does not fully meet PLOS ONE’s publication criteria as it currently stands. Therefore, we invite you to submit a revised version of the manuscript that addresses the points raised during the review process.

ACADEMIC EDITOR: Please revise the manuscript according to the reviewer comments.A major revision is required..==============================

We look forward to receiving your revised manuscript.

Kind regards,

Abdelazeem Mohamed Algammal, Prof, Ph.D

Academic Editor

PLOS ONE

Journal Requirements:

"This research was funded by the Antimicrobial Resistance Cross Council Initiative supported by the seven research councils in partnership with the Department of Health and Department for Environment Food & Rural Affairs (NERC/ BBSRC/MRC grant number: NE/N019555/1). "

"This research was funded by the Antimicrobial Resistance Cross Council Initiative supported by the seven research councils in partnership with the Department of Health and Department for Environment Food & Rural Affairs (NERC/ BBSRC/MRC grant number: NE/N019555/1). Dr. Emily Rousham received this grant."

4. Please amend the manuscript submission data (via Edit Submission) to include author Homaun Kabir Chowdhury.

Reviewers' comments:

Reviewer's Responses to Questions

**Comments to the Author**

1. Is the manuscript technically sound, and do the data support the conclusions?

Reviewer #1: Partly

Reviewer #2: Yes

2. Has the statistical analysis been performed appropriately and rigorously? 

Reviewer #1: Yes

Reviewer #2: Yes

3. Have the authors made all data underlying the findings in their manuscript fully available?

Reviewer #1: No

Reviewer #2: Yes

4. Is the manuscript presented in an intelligible fashion and written in standard English?

Reviewer #1: No

Reviewer #2: Yes

5. Review Comments to the Author

Reviewer #1: Comments to authors:

- The current study is interesting; however, the authors should address the following comments to improve the quality of the manuscript:

- The manuscript should be revised for language editing and grammar mistakes.

Title:

I think the work would benefit from the title that contains the main conclusion of the study (should be derived from the conclusion). Please modify the title.

Abstract:

- The abstract must illustrate the used methods and the most prevalent results (give more hints about methods and results). Besides, rephrase the main conclusion of your findings.

Introduction: (it needs to be more informative)

-Give a hint about different infection caused by E. coli, virulence factors, and the mechanism of disease occurrence.

- The authors should illustrate the public health importance concerning the emergence of multidrug-resistant (MDR) bacterial pathogens that reflecting the necessity of new potent and safe antimicrobial agents. Several studies proved the widespread MDR- bacterial pathogens;

Authors could add the following paragraph:

Multidrug resistance has been increased all over the world that is considered a public health threat. Several recent investigations reported the emergence of multidrug-resistant bacterial pathogens from different origins including humans, birds, cattle, and fish that increase the need for routine application of the antimicrobial susceptibility testing to detect the antibiotic of choice as well as the screening of the emerging MDR strains. You should cite the following valuable studies:

1.PMID: 33177849

2.PMID: 32497922

3.PMID:33061472

4.PMID: 33947875

5.PMID: 32472209

6.PMID: 31170450

7.PMID: 33188216

8. Abouelmaatti, R. et al. (2013): Cloning and analysis of Nile tilapia Toll-like receptors type-3 mRNA: Centr. Eur. J. Immunol; 38 (3): 277-282. DOI: https://doi.org/10.5114/ceji.2013.3774020

9.PMID: 30150182

10. PMID: 34445951

-Rephrase the aim of the work to be clear and better sound.

Material and methods

-Add the following title to the Methods section:

Isolation and identification of E. coli:

• Discuss in detail the methods of isolation and identification of E. coli. Besides, specific references should be added.

•Add the company, city, and country of the used bacterial media and reagents that were used in the biochemical identification of isolates. Also, enumerate all used biochemical reactions.

- Illustrate in a new table primers sequences and cycling conditions of PCR based detection of plasmid mediated quinolone resistance genes

- Antimicrobial susceptibility testing:

•Illustrate the antimicrobial classes of the tested antimicrobial agents.

•The authors are advised to classify the tested isolates to MDR or XDR as described by Magiorakos et al.

Magiorakos AP, Srinivasan A, Carey RB, Carmeli Y, Falagas ME, Giske CG, et al. Multidrug-resistant, extensively drug-resistant and pandrug-resistant bacteria: An international expert proposal for interim standard definitions for acquired resistance. Clin Microbiol Infect. 2012; 18:268–81. doi:10.1111/j.1469-0691.2011.03570.x.

- Where are the accession numbers of the sequenced PMQR genes????

-Add more data about the used software in the statistical analyses?

-Result:

-Illustrate the phenotypic characteristics of the recovered -Illustrate the phenotypic characteristics of E. coli.

-Illustrate in a new table the occurrence of MDR (Multidrug resistance) among the recovered isolates (illustrate the names of the antimicrobial classes and different antibiotics):

No. of strains%Type of resistance

R OR MDR OR XDRPhenotypic multidrug resistance

(Antimicrobial classes and different antibiotics).

The antibiotic -resistance genes

- Where are the figures of the phylogenetic analysis??

-Discussion:

- The authors are advised to illustrate the real impact of their findings without repetition of results.

-Illustrate the different mechanisms of antimicrobial resistance in E. coli.

-Conclusion

- Should be rephrased to be sounded. A real conclusion should focus on the question or claim you articulated in your study, which resolution has been the main objective of your paper?

Reviewer #2: - The current study has a significant impact, but it needs a major revision:

- The manuscript should be revised for grammar mistakes.

- Please write the scientific names of bacterial pathogens and genes in the correct form all over the manuscript and in the References section (should be italic).

-The title is broad, please modify the title.

- Add more details about the used methods and most prevalent results in the abstract.

-In the introduction: discuss the public health importance of the E. coli and different infections caused by them.

-Improve the aim of work.

Methods:

-Explain the methods of isolation and identification in detail??

-Specific references should be added to all the used methods and techniques.

-Add the manufacturing company, city, and country for the used reagents and antimicrobial discs.

- Add the accession numbers of the sequenced PMQR genes (necessary).

--Results:

- Discuss in detail the phenotypic characters of the isolated E. coli strains.

- Increase the resolutions of all figures in the main manuscript.

-You must support your results with the figures of the phylogenetic analysis.

-Discussion:

- Please improve (Avoid repetition of results)

-Please improve the main conclusion of the manuscript.

6. PLOS authors have the option to publish the peer review history of their article (what does this mean?). If published, this will include your full peer review and any attached files.

Reviewer #1: No

Reviewer #2: No

---

## [Author Response · Author response to Decision Letter 0]

12 Dec 2021

Response: We have revised and formatted accordingly now. 

Response: We have corrected the funding information and award number in Funding information.

"This research was funded by the Antimicrobial Resistance Cross Council Initiative supported by the seven research councils in partnership with the Department of Health and Department for Environment Food & Rural Affairs (NERC/ BBSRC/MRC grant number: NE/N019555/1). "

"This research was funded by the Antimicrobial Resistance Cross Council Initiative supported by the seven research councils in partnership with the Department of Health and Department for Environment Food & Rural Affairs (NERC/ BBSRC/MRC grant number: NE/N019555/1). Dr. Emily Rousham received this grant."

Response: We have removed the funding information from the manuscript and included the funding statement in the cover letter as follows:

This research was funded by the Antimicrobial Resistance Cross Council Initiative supported by the seven research councils in partnership with the Department of Health and Department for Environment Food & Rural Affairs (NERC/ BBSRC/MRC grant number: NE/N019555/1). 

4. Please amend the manuscript submission data (via Edit Submission) to include author Homaun Kabir Chowdhury.

Response: We have incorporated the author via Edit Submission.

Response: We have revised the abstract in the manuscript and amended the abstract via Edit Submission to make them identical.

Response: We have moved the ethics statement in the method section of the manuscript. (Line: 118-120)

Reviewers' comments:

Reviewer's Responses to Questions

Comments to the Author

1. Is the manuscript technically sound, and do the data support the conclusions?

Reviewer #1: Partly

Reviewer #2: Yes

Response: Thank you for your appreciation.________________________________________

2. Has the statistical analysis been performed appropriately and rigorously?

Reviewer #1: Yes

Reviewer #2: Yes

Response: Thank you for the comments.________________________________________

3. Have the authors made all data underlying the findings in their manuscript fully available?

Reviewer #1: No

Reviewer #2: Yes

Response: We have included all the data in the manuscript and also in data availability statement in detail. ________________________________________

4. Is the manuscript presented in an intelligible fashion and written in standard English?

Reviewer #1: No

Reviewer #2: Yes

Response: We have revised the entire manuscript including data presentation, English correction according to your comments.________________________________________

5. Review Comments to the Author

Reviewer #1: Comments to authors:

- The current study is interesting; however, the authors should address the following comments to improve the quality of the manuscript:

Response: Thank you for your comment on our manuscript.

- The manuscript should be revised for language editing and grammar mistakes.

Response: we have revised the entire manuscript for English by our co-author, Dr. Emily Rousham who is a British professor at Loughborough University, UK. 

Title:

I think the work would benefit from the title that contains the main conclusion of the study (should be derived from the conclusion). Please modify the title.

Response: We have modified the titles highlighting our main results. (Line:1-3)

Abstract:

- The abstract must illustrate the used methods and the most prevalent results (give more hints about methods and results). Besides, rephrase the main conclusion of your findings.

Response: We have expanded the method section in the abstract and also incorporated more results. Finally, we rephrased the conclusion in abstract. (Line:19-50) 

Introduction: (it needs to be more informative)

-Give a hint about different infection caused by E. coli, virulence factors, and the mechanism of disease occurrence.

Response: Thank for your comments. We have included information about diseases caused by E. coli and its severity when E. coli becomes multi-drug resistant in nature. (Line:58-63;)

- The authors should illustrate the public health importance concerning the emergence of multidrug-resistant (MDR) bacterial pathogens that reflecting the necessity of new potent and safe antimicrobial agents. Several studies proved the widespread MDR- bacterial pathogens;

Authors could add the following paragraph:

Multidrug resistance has been increased all over the world that is considered a public health threat. Several recent investigations reported the emergence of multidrug-resistant bacterial pathogens from different origins including humans, birds, cattle, and fish that increase the need for routine application of the antimicrobial susceptibility testing to detect the antibiotic of choice as well as the screening of the emerging MDR strains. You should cite the following valuable studies:

1.PMID: 33177849

2.PMID: 32497922

3.PMID:33061472

4.PMID: 33947875

5.PMID: 32472209

6.PMID: 31170450

7.PMID: 33188216

8. Abouelmaatti, R. et al. (2013): Cloning and analysis of Nile tilapia Toll-like receptors type-3 mRNA: Centr. Eur. J. Immunol; 38 (3): 277-282. DOI: https://doi.org/10.5114/ceji.2013.3774020

9.PMID: 30150182

10. PMID: 34445951

Response: Thank you for the important references. We have cited most of these publications. (Line:63-66) 

-Rephrase the aim of the work to be clear and better sound.

Response: We rephrased the aim of the study according to your suggestion. (Line:111-116) 

Material and methods

-Add the following title to the Methods section:

Isolation and identification of E. coli:

• Discuss in detail the methods of isolation and identification of E. coli. Besides, specific references should be added.

Response: We have added this heading to the method section. (Line:135-149).

•Add the company, city, and country of the used bacterial media and reagents that were used in the biochemical identification of isolates. Also, enumerate all used biochemical reactions.

Response: Thank you for your comment. We have added company, city and country name for media, reagents and equipment used in this study. (Line:137-138, 143, 148-149, 160-162, 171, 189, 205-206) 

- Illustrate in a new table primers sequences and cycling conditions of PCR based detection of plasmid mediated quinolone resistance genes

Response: A new table entitled “List of Primer sequences used in multiplex PCR for the determination of PMQR determinants” has been inserted. (Line:155)

- Antimicrobial susceptibility testing:

•Illustrate the antimicrobial classes of the tested antimicrobial agents.

•The authors are advised to classify the tested isolates to MDR or XDR as described by Magiorakos et al.

Magiorakos AP, Srinivasan A, Carey RB, Carmeli Y, Falagas ME, Giske CG, et al. Multidrug-resistant, extensively drug-resistant and pandrug-resistant bacteria: An international expert proposal for interim standard definitions for acquired resistance. Clin Microbiol Infect. 2012; 18:268–81. doi:10.1111/j.1469-0691.2011.03570.x.

Response: We have classified the tested E. coli isolates according to Magiorakos et al. and described susceptibility patterns against different classes of antibiotics in a new table entitled “Occurrence of clinically important antibiotic resistance among PMQR positive E. coli isolates from aquatic environments in Bangladesh”. (Line:175-179; 235-245)

- Where are the accession numbers of the sequenced PMQR genes????

Response: We have submitted the sequences to the GeneBank and received the accession numbers: qnrS (OL439745), qnrB (OL439744), oqxAB (OK668389), qepA (OK668390) and aac(6’)-lb-cr (OL439743). (Line:). However, the numbers are not yet released in the web as it takes time after processing by GeneBank team. (Line:165-167) 

-Add more data about the used software in the statistical analyses?

Response: We have revised the statistical analysis section with more information. (Line:189-197)

-Result:

-Illustrate the phenotypic characteristics of the recovered -Illustrate the phenotypic characteristics of E. coli.

Response: We have described the phenotypic characteristics of E. coli isolates in result section as follows (Line:211-219):

Isolates were identified as E. coli based on their morphology and biochemical characteristics. Microscopically, the bacteria appeared as Gram-negative moderate size, motile, and non-sporulated rods. The bacteria grew well on mTEC agar and appeared as red/magenta color colonies due to β-D-glucuronidase activity which is highly specific for E. coli. On MacConkey agar, bacteria produced characteristic pink colonies due to lactose fermentation. Biochemically, all isolates were positive for catalase, lactose fermentation, indole and methyl-red, tests. Simultaneously, they were negative for cytochrome oxidase, Voges-Proskauer, citrate-utilization, H2S production, and urease tests. All 300 water samples were positive for E. coli. 

 -Illustrate in a new table the occurrence of MDR (Multidrug resistance) among the recovered isolates (illustrate the names of the antimicrobial classes and different antibiotics):

No. of strains%Type of resistance

R OR MDR OR XDR Phenotypic multidrug resistance

(Antimicrobial classes and different antibiotics).

The antibiotic -resistance genes

Response: We have inserted a new table entitled “Occurrence of clinically important antibiotic resistance among PMQR positive E. coli isolates from aquatic environments in Bangladesh” where data on MDR and XDR strains %, resistance % against different antibiotic classes were included according to Magiorakos et al. (Line:235-245) 

- Where are the figures of the phylogenetic analysis??

Response: We have analysed phylogenetic relatedness of the PMQR positive isolates using ERIC-PCR and generated a dendrogram to determine their clonal relationships. (Line:277-290)

-Discussion:

- The authors are advised to illustrate the real impact of their findings without repetition of results.

Response: We appreciate this comment. We have revised the discussion section substantially according to your comment. (Line:292-355).

-Illustrate the different mechanisms of antimicrobial resistance in E. coli.

Response: we have briefly described the different resistance mechanisms in E. coli in discussion section. (Line:319-325).

-Conclusion

- Should be rephrased to be sounded. A real conclusion should focus on the question or claim you articulated in your study, which resolution has been the main objective of your paper?

Response: We thank the reviewer for this comment. We revamped the conclusion section to address the reviewers concern (lines:357-363). The new conclusion read as follows: 

Our study shows a high prevalence of PMQR-positive E. coli in urban and rural waters. These plasmid-mediated isolates were mostly MDR, predominantly ESBL-producing and genetically heterogeneous. The high prevalence of plasmid-mediated quinolone resistance poses a risk for horizontal gene transfer and this, in association with ESBL genes, adds to the threat of AMR transmission via the environment. This study highlights the importance of including surface waters and wastewaters in One Health AMR surveillance programs to understand the emergence and transmission dynamics of AMR and for designing environmental intervention strategies. 

Reviewer #2: - The current study has a significant impact, but it needs a major revision:

- The manuscript should be revised for grammar mistakes.

Response: Thank you for your comment. We have revised the entire manuscript and checked English grammar carefully. 

- Please write the scientific names of bacterial pathogens and genes in the correct form all over the manuscript and in the References section (should be italic).

Response: We have correctly written scientific names of all pathogens and genes throughout the manuscript including reference section. 

-The title is broad, please modify the title.

Response: We have modified the title and made it specific. (Line:1-3)

- Add more details about the used methods and most prevalent results in the abstract.

Response: We have elaborated the methods and most prevalent results in the abstract. (Line:19-50) 

-In the introduction: discuss the public health importance of the E. coli and different infections caused by them.

Response: We have described the public health importance of E. coli and infections caused by this organism. (Line:53-57,59-63)

-Improve the aim of work.

Response: We have revised the aim of the work. (Line:111-116)

Methods:

-Explain the methods of isolation and identification in detail??

Response: We have included a separate section on the isolation and identification of E. coli in the method section. (Line:135-149)

-Specific references should be added to all the used methods and techniques.

Response: We have added references to all the used methods and techniques. (Line:147)

-Add the manufacturing company, city, and country for the used reagents and antimicrobial discs. 

Response: We have provided company, city and country information for all the reagents, media and equipment used in the study. (Line:137-138, 143, 148-149, 160-162, 171, 189, 205-206)

- Add the accession numbers of the sequenced PMQR genes (necessary).

Response: We have submitted the sequences to the GeneBank and received the accession numbers: qnrS (OL439745), qnrB (OL439744), oqxAB (OK668389), qepA (OK668390) and aac(6’)-lb-cr (OL439743). (Line:). However, the numbers are not yet released in the web as it takes time after processing by GeneBank team. (Line:165-167)

--Results:

- Discuss in detail the phenotypic characters of the isolated E. coli strains.

Response: We have described the phenotypic characteristics of E. coli isolates in result section in Line 211-219 as follows:

Isolates were identified as E. coli based on their morphology and biochemical characteristics. Microscopically, the bacteria appeared as Gram-negative moderate size, motile, and non-sporulated rods. The bacteria grew well on mTEC agar and appeared as red/magenta colour colonies due to β-D-glucuronidase activity which is highly specific for E. coli. On MacConkey agar, bacteria produced characteristic pink colonies due to lactose fermentation. Biochemically, all isolates were positive for catalase, lactose fermentation, indole and methyl-red, tests. Simultaneously, they were negative for cytochrome oxidase, Voges-Proskauer, citrate-utilization, H2S production, and urease tests. All 300 water samples were positive for E. coli. 

- Increase the resolutions of all figures in the main manuscript.

Response: we have increased the resolution (300dpi) of all figures used in this study. 

-You must support your results with the figures of the phylogenetic analysis.

Response: We have analysed phylogenetic relatedness of the PMQR positive isolates using ERIC-PCR and generated a dendrogram to determine their clonal relationships. (Line:277-290) 

-Discussion:

- Please improve (Avoid repetition of results)

Response: we have revised the discussion section substantially according to your comment. (Line:292-355) 

-Please improve the main conclusion of the manuscript.

Response: We have modified the conclusion of the manuscript in Line:357-363 as follows:

Our study shows a high prevalence of PMQR-positive E. coli in urban and rural waters. These plasmid-mediated isolates were mostly MDR, predominantly ESBL-producing and genetically heterogeneous. The high prevalence of plasmid-mediated quinolone resistance poses a risk for horizontal gene transfer and this, in association with ESBL genes, adds to the threat of AMR transmission via the environment. This study highlights the importance of including surface waters and wastewaters in One Health AMR surveillance programs to understand the emergence and transmission dynamics of AMR and for designing environmental intervention strategies. ________________________________________

---

## [Decision Letter · Decision Letter 1]

15 Dec 2021

High prevalence of plasmid-mediated quinolone resistance (PMQR) among E. coli from aquatic environments in Bangladesh

PONE-D-21-32725R1

Dear Dr. Amin,

We’re pleased to inform you that your manuscript has been judged scientifically suitable for publication and will be formally accepted for publication once it meets all outstanding technical requirements.

Kind regards,

Abdelazeem Mohamed Algammal, Prof, Ph.D

Academic Editor

PLOS ONE

Additional Editor Comments (optional):

Reviewers' comments:

Reviewer's Responses to Questions

**Comments to the Author**

1. If the authors have adequately addressed your comments raised in a previous round of review and you feel that this manuscript is now acceptable for publication, you may indicate that here to bypass the “Comments to the Author” section, enter your conflict of interest statement in the “Confidential to Editor” section, and submit your "Accept" recommendation.

Reviewer #1: All comments have been addressed

2. Is the manuscript technically sound, and do the data support the conclusions?

Reviewer #1: Yes

3. Has the statistical analysis been performed appropriately and rigorously? 

Reviewer #1: Yes

4. Have the authors made all data underlying the findings in their manuscript fully available?

Reviewer #1: Yes

5. Is the manuscript presented in an intelligible fashion and written in standard English?

Reviewer #1: Yes

6. Review Comments to the Author

Reviewer #1: The authors have carried out significant changes to the manuscript. They have addressed all the suggested corrections and comments. Really, it's an interesting study that has a significant impact. Now, the manuscript could be accepted.

7. PLOS authors have the option to publish the peer review history of their article (what does this mean?). If published, this will include your full peer review and any attached files.

Reviewer #1: No

---

## [Editor Report · Acceptance letter]

19 Dec 2021

PONE-D-21-32725R1 

High prevalence of plasmid-mediated quinolone resistance (PMQR) among *E. coli* from aquatic environments in Bangladesh 

Dear Dr. Amin:

I'm pleased to inform you that your manuscript has been deemed suitable for publication in PLOS ONE. Congratulations! Your manuscript is now with our production department. 

Kind regards, 

on behalf of

Professor Abdelazeem Mohamed Algammal 

Academic Editor

PLOS ONE